# Cosmic Whirl: Navigating the Comet Trail in DNA: H2AX Phosphorylation and the Enigma of Uncertain Significance Variants

**DOI:** 10.3390/genes15060724

**Published:** 2024-06-01

**Authors:** Sevdican Ustun Yilmaz, Nihat Bugra Agaoglu, Karin Manto, Meltem Muftuoglu, Ugur Özbek

**Affiliations:** 1Department of Medical Biotechnology, Institute of Health Sciences, Acibadem Mehmet Ali Aydinlar University, 34752 Istanbul, Türkiye; sevdican.yilmaz@live.acibadem.edu.tr (S.U.Y.); meltem.muftuoglu@acibadem.edu.tr (M.M.); 2Department of Medical Genetics, Umraniye Training and Research Hospital, University of Health Sciences, 34764 Istanbul, Türkiye; nihatbugra.agaoglu@ikf-khnw.de; 3IKF-The Frankfurt Institute of Clinical Cancer Research, 60488 Frankfurt am Main, Germany; 4Department of Genome Studies, Institute of Health Sciences, Acibadem Mehmet Ali Aydinlar University, 34752 Istanbul, Türkiye; karin.manto@live.acibadem.edu.tr; 5Department of Molecular Biology and Genetics, Faculty of Engineering and Natural Sciences, Acibadem Mehmet Ali Aydinlar University, 34752 Istanbul, Türkiye; 6Izmir Biomedicine and Genome Center (IBG), 35340 Izmir, Türkiye

**Keywords:** hereditary breast and ovarian cancer (HBOC) syndrome, *BRCA2*, γH2AX, comet, VUS classification, DNA Damage, functional assays

## Abstract

Pathogenic variations in the *BRCA2* gene have been detected with the development of next-generation sequencing (NGS)-based hereditary cancer panel testing technology. It also reveals an increasing number of variants of uncertain significance (VUSs). Well-established functional tests are crucial to accurately reclassifying VUSs for effective diagnosis and treatment. We retrospectively analyzed the multi-gene cancer panel results of 922 individuals and performed in silico analysis following ClinVar classification. Then, we selected five breast cancer-diagnosed patients’ missense *BRCA2* VUSs (T1011R, T1104P/M1168K, R2027K, G2044A, and D2819) for reclassification. The effects of VUSs on BRCA2 function were analyzed using comet and H2AX phosphorylation (γH2AX) assays before and after the treatment of peripheral blood mononuclear cells (PBMCs) of subjects with the double-strand break (DSB) agent doxorubicin (Dox). Before and after Dox-induction, the amount of DNA in the comet tails was similar in VUS carriers; however, notable variations in γH2AX were observed, and according to combined computational and functional analyses, we reclassified T1001R as VUS-intermediate, T1104P/M1168K and D2819V as VUS (+), and R2027K and G2044A as likely benign. These findings highlight the importance of the variability of VUSs in response to DNA damage before and after Dox-induction and suggest that further investigation is needed to understand the underlying mechanisms.

## 1. Introduction

Breast cancer (BC) and ovarian cancer (OC) occur with a combination of environmental and genetic risk factors and appear almost 10–30% more prevalently within families [1]. Only 10% of these cases are hereditary and related to a specific genetic mutation in a cancer predisposition gene that may be passed down through generations [2]. Hereditary Breast and Ovarian Cancer (HBOC) is a genetic condition that raises an individual’s risk of developing BC, particularly before the age of 50, and OC [3]. Onco-suppressors BC gene 1 (*BRCA1_*OMIM 113705) and BC gene 2 (*BRCA2*_OMIM 600185) are the main genes in HBOC syndrome and play a role in response to DNA damage, and their pathogenic variant carriers account for around 30–50% of cases having a higher risk of HBOC [4]. The general population’s BC and OC risk is 12% and 1–2%, and by age 70, this increases with the germline pathogenic mutations of *BRCA2* 45–69% and 11–17%, respectively [5].

Therefore, next-generation sequencing (NGS) technology in hereditary cancers uses targeted diagnostic panels to find several genomic variants in cancer susceptibility genes [6]. This knowledge helps diagnose the disease, monitor its progression, devise strategies to reduce the risk, and recommend appropriate prophylactic procedures for patients and their family members [7]. In addition, evaluating the affected gene(s) can provide valuable insight into individualized therapeutic options or preventive surgical alternatives [8,9]. In this context, the accurate and precise interpretation of these variants is very important for the management of hereditary cancer syndrome, and as more patients undergo NGS panel testing, the number of variants identified is gradually increasing [10]. 

For these reasons, to facilitate rapid and efficient clinical management, the American College of Medical Genetics (ACMG) and the Association for Molecular Pathology (AMP) released guidelines in 2015 to standardize variant interpretations [11]. These criteria consider population-based, computational, predictive, functional, allelic, and familial segregation data, along with the parental origin of the variant. Based on this, the variant is grouped into one of the following categories: pathogenic (P), likely pathogenic (LP), variants of uncertain significance (VUS), likely benign (LB), and benign (B) [11]. Only the interpretation of germline variants whose functional significance or clinical relevance is uncertain, frequently referred to as VUSs, could be complicated between these groups. A well-established functional assay is necessary to accurately classify missense uncertain BRCA2 variants as either P/LP or B/LB with the strong evidence codes PS3 and BS3, respectively [12], so that clinicians can better advise patients on cancer risk and treatment alternatives. 

According to the March 2024 data from the ClinVar database, there are 18025 variants in the *BRCA2* gene (NM_000059.4). Among these variants, 7736 are classified as single-nucleotide variations, their molecular consequences were missense, and their germline classification was uncertain (https://www.ncbi.nlm.nih.gov/clinvar, accessed on 4 March 2024) [13]. 

Multiple genetic and biochemical functional tests have been published, demonstrating BRCA2’s role in DNA damage by regulating the homologous recombination (HR) repair pathway and interacting with DNA repair proteins such as phosphorylated histone H2AX (γH2AX) and RAD51 [14,15]. According to standards for the quantitative measurement of DNA damage, DNA double-strand breaks (DSBs) created by anti-tumor drugs are measured by single-cell alkaline gel electrophoresis (comet) and γH2AX protein expression [16]. Doxorubicin (Dox), a potent anthracycline antibiotic, has multiple anticancer activities, including DSBs, reactive oxygen species (ROS) production, apoptosis, senescence, autophagy, ferroptosis, and pyroptosis, as well as the ability to modulate the immune system [17]. 

In this study, we wanted to assess how missense *BRCA2* VUSs can cause DNA damage before and after the DSB-inducer chemotherapeutic drug doxorubicin. This might help us to estimate BRCA2 variants’ pathogenicity. For this purpose, the study participants’ peripheral blood mononuclear cells (PBMCs) were obtained to determine the percentage of damaged DNA in the tail using the comet assay and to measure the expression level of γH2AX. Subsequently, we combined the in silico results with functional assay results and reclassified the *BRCA2* VUSs using the ACMG/AMP variant classification framework.

## 2. Materials and Methods

### 2.1. The Subjects and Selection of Missense VUSs in the BRCA2 Gene

We retrospectively analyzed the NGS-based hereditary cancer panel results of 922 index cases or the cases that were tested due to a positive family history, who were referred to the Hereditary Cancer Clinic in the Genomic Laboratory (GLAB) of Umraniye Training and Research Hospital [18] and Medical Genetics Department of Acibadem Maslak Hospital, Istanbul, Türkiye, between November 2017 and June 2022, and filtered their results, as provided in Appendix A. 

The present study was approved by the Acibadem Mehmet Ali Aydinlar University (ACU) and Acibadem Healthcare Institutions Medical Research Ethics Committee (ATADEK) (approval no. 2023-10/437) and was performed in accordance with the guidelines of human research. All subjects involved in the study were informed about the study and provided their written informed consent according to the Declaration of Helsinki, as approved by the Ethics Committee of ACU and ATADEK. All personal data were anonymized, and the samples were coded.

### 2.2. Data Collection about Variants

We selected missense *BRCA2* VUSs, described in Figure 1 and Table 1, and available information was collected from ClinVar [13] and other databases, as provided in Appendix A.

### 2.3. PBMC Isolation

PBMCs were separated from whole blood, as provided in Appendix A.

### 2.4. Comet Assay

Comet assays were performed under alkaline conditions using the CometAssay^®^ Kit (Trevigen, Gaithersburg, MA, USA) according to the manufacturer’s instructions and as described in Ogulur et al. (2021) [19]. Details are provided in Appendix A. 

### 2.5. Western Blot γH2AX Assay

Protein lysates derived from lymphocytes were resolved on SDS-PAGE gels and transferred to polyvinylidene-fluoride (PVDF) membranes, as provided in Appendix A.

### 2.6. ACMG/AMP Evidence-Based Reclassification of Variants

We have conducted a retrospective assessment and reclassification of missense *BRCA2* VUSs utilizing ACMG/AMP rules and incorporating updated information on the variations. According to in silico tools, variants were interpreted with PP3 and BP4 criteria for pathogenicity or benignity, respectively [20]. Additionally, in this study, VUSs were evaluated as VUS (+) close to LP, VUS (−) close to LB, and VUS-intermediate (int) as the average value. According to functional analysis, variants were interpreted using PS3 and BS3 criteria for pathogenicity or benignity, respectively [12]. Based on this, variants were grouped into one of the following categories: P, LP, VUS (+), VUS (int), VUS (−), LB, and B.

### 2.7. Statistical Analysis

All graphs were obtained for the samples according to their genetic status: non-carrier (NC), pathogenic variant carrier (PC), or VUS-Cs. Statistical analysis and graphs were performed using GraphPad Prism 9 (GraphPad InStat). The statistical significance level is 0.05. Details are provided in Appendix A. 

## 3. Results

### 3.1. Clinical Consequences of Variants Detected with Hereditary Cancer Panel Testing

We recently retrospectively evaluated the hereditary cancer panel results of the cases with a history of HBOC. We obtained 89 different *BRCA2* variants from 107 cases (Appendix A). Of these variants, 24 are LP/P, 14 are LB/B, 43 are CIP or VUS, and 8 are NR, according to the ClinVar database. We filtered the cohort to establish a functional study for *BRCA2* SNV missense VUSs. We had 41 missense SNVs among the 51 *BRCA2* VUS/CIP/NR variants in the cohort. To proceed with the study, we obtained clinical reports from 45 patients with the 41 identified variants from our partner hospitals (Appendix A). We contacted female cases just with a history of HBOC and excluded others from the study for the reasons as provided in Appendix A. 

We selected the following five breast cancer-diagnosed patients’ missense *BRCA2* (NM_000059.4, NP_000050.3) VUSs for reclassification by evaluating their DNA damage activities with functional analyses: VUS-C1 has *BRCA2* c.3032C>G, p.Thr1011Arg (T1011R), (rs80358548), VUS-C2 has two *BRCA2* variants; c.3310A>C, p.Thr1104Pro (T1104P), (rs80358577) and c.3503T>A, p.Met1168Lys (M1168K), (rs80358598), and a nonsense *ATM* (NM_000051.4, NP_000042.3) c.4473C>T, p.Phe1491= (F1491=), (rs4988008), VUS-C3 has *BRCA2* c.6080G>A, p.Arg2027Lys (R2027K), (rs431825337), VUS-C4 has *BRCA2* c.6131G>C, p.Gly2044Ala (G2044A), (rs56191579), VUS-C5 has *BRCA2* c.8456A>T, p.Asp2819Val (D2819V), (rs1555287655). We also select a PC for the study to validate the sensitivity of the functional assay as a positive control that carries a pathogenic *BRCA1* (NM_007294.4, NP_009225.1) c.2131_2132del, p.Lys711fs (frameshift K711fs) deletion, (rs398122653), a *BRCA2* missense VUS c.4277C>T, p.Thr1426Ile (T1426I), (rs748591104) and an *APC* c.3449G>C, p.Glu1317Gln (E1317Q), (rs1801166) (Figure 1, Table 1). Thus, five missense *BRCA2* VUS-Cs (T1011R, T1104P/M1168K, R2027K, G2044A, D2819V) and one PC breast cancer-diagnosed female cases participated in this study. We also included two healthy NC participants as negative controls and chose them from the same cohort who have a family with a history of HBOC. In NCs, we did not detect any VUSs by using NGS with the same hereditary cancer panel. The women’s personal and clinical information and the familial history of each case are shown in Appendix A.

### 3.2. In Silico Analyses after ClinVar Classification

We analyzed the population allele frequencies of these seven variants by using the gnomAD database to determine their global prevalence. Our findings revealed that two variants (R2027K and D2819V) were not previously reported, and the remaining variants (T1426I, T1011R, T1104P/M1168K, G2044A) were found in low frequency in the global population (Table 1). These results provide moderate evidence of their potential pathogenicity (PM2) level [11]. However, the ClinGen Sequencing Variant Interpretation (SVI) Working Group suggested that the result should be reported as benign, and the moderate evidence should be changed to supporting evidence [21]. 

As shown in Figure 1, the T1011R, T1104P/M1168K, T1426I, R2027K, and G2044A variants are in the RAD51-binding domain and in exon 11, which is a coldspot region on the protein where missense variants are unexpected to be pathogenic, and D2819V is in the DNA-binding domain and exon 19. 

VUS-C1, T1011R, is in the BRCA2 repeat domain, and computational analyses (SIFT = 0.0, PolyPhen-2 = 1.0, MetaRNN = 0.67, DeMAG = 0.76) predict that this variant has a damaged effect on the protein. This sequence change replaces threonine, which is neutral and polar, with arginine, which is basic and polar, at codon 1011 of the BRCA2 protein. VUS-C2, T1104P, computational analyses (SIFT = 0.0, PolyPhen-2 = 1, MetaRNN = 0.65, DeMAG = 0.64) predict that this variant is deleterious. Threonine at codon 1104 is highly conserved; threonine is a polar amino acid, while proline is a nonpolar amino acid with unique structural properties. The substitution of threonine with proline at position 1104 may lead to alterations in the protein’s three-dimensional structure and potentially affect its interactions with other proteins or DNA. However, combined evidence from multiple other in silico predictors determines benign supporting evidence for this variant. In addition, databases reported this variant in seven patients together with BRCA2 M1168K. This is the same as VUS-C2 in our cohort and proposing the “in cis” phase for two of them. VUS-C2, M1168K, is predicted to be benign with computational analysis tools (SIFT = 0.16, PolyPhen-2 = 0.006, MetaRNN = 0.30, DeMAG = 0.08). Substituting methionine with lysine at position 1168 may have functional implications for the BRCA2 protein. Methionine is a hydrophobic amino acid, while lysine is a positively charged amino acid with a hydrophilic side chain. This change in amino acid properties could affect the protein’s structure, stability, or interactions with other molecules. VUS-C3, R2027K, computational analyses (SIFT = 0.49, PolyPhen-2 = 0.009, MetaRNN = 0.11, DeMAG = 0.06) predict a benign effect of the variant on protein function. Despite this, the substitution of arginine with lysine at position 2027 may potentially affect the function of the BRCA2 protein. Arginine is a positively charged amino acid, whereas lysine is positively charged but differs in its side chain structure. This change in charge and side chain properties could disrupt protein–protein interactions or alter the protein’s structural conformation. VUS-C4, G2044A, computational analyses (SIFT = 0.7, PolyPhen-2 = 0.062, MetaRNN = 0.11, DeMAG = 0.06) predict a benign effect of the variant on protein function. The glycine-to-alanine substitution may or may not significantly affect the protein’s structure, stability, or function. VUS-C5, D2819V, located in the OB-2 domain of the BRCA2 protein, and computational analyses (SIFT = 0.0, PolyPhen-2 = 0.964, MetaRNN = 0.84, DeMAG = 0.62) predict the damaged effect on the protein. The substitution of aspartic acid with valine at position 2819 may potentially impact the structure and function of the BRCA2 protein. Aspartic acid is an acidic amino acid with a negatively charged side chain, while valine is a nonpolar amino acid with a hydrophobic side chain. This change in amino acid properties could disrupt protein–protein interactions, folding, or stability.

According to the results of the in silico analysis, the variants were preclassified before the functional analysis with the ACMG/AMP evidence code PP3/BP4. We added PP3 evidence code to the variants, which has a deleterious effect on the gene and classified T1011R as VUS(−), T1104P as VUS(int)/M1168K VUS(−), and D2819V as VUS(int). However, R2027K and G2044A were interpreted as likely benign with the BP4 evidence code (Table 1). 

### 3.3. Dox-Induction Increased DNA Damage among NCs

At the beginning of the study, we examined Dox-induction’s effect on NC cells. We revealed a noticeable augmentation in DNA damage after administering 0.5 μM Dox for an hour (Figure 2). 

The NC group showed a statistically significant increase of 1.6-fold in the mean DNA percentage in the comet tail (20.05 ± 3.5 vs. 32.16 ± 5.5, *p* = 0.0138) (Figure 3A). Dox-induction increased γH2AX levels 1.37-fold (0.24 ± 0.04 vs. 0.33 ± 0.07), but this increase was not statistically significant (*p* > 0.05) (Figure 3B). The demonstrated damaging effects caused by the drug on non-pathological subjects confirm the appropriateness of the dosage and duration employed in our investigations.

### 3.4. Endogenous DNA Damage Levels Are Considerably Higher in PC

We analyzed DNA damage levels by dividing participants into three groups: two healthy NCs, one PC, and five VUS-Cs. The % DNA in tail results revealed that the PC group (33.98 ± 6.7) had a 1.7-fold higher endogenous DNA damage level than the NC (20.05 ± 3.5, *p* > 0.05) and VUS (18.83 ± 3.6, *p* > 0.05) groups (Figure 3A). After Dox-induction, the mean % tail DNA of the NC group (32.16 ± 5) showed a 2-fold and 2.5-fold increase compared to the PC (16.97 ± 2.1, *p* = 0.0088) and VUS-C (13.08 ± 1.8, *p* = 0.0051) groups, respectively (Figure 3A). Comparing the alteration of each group’s endogenous and exogenous levels, the NC group increased 1.6-fold (20.05 ± 3.5 to 32.16 ± 5.5, *p* = 0.0138) (Figure 3A). The PC decreased 0.5-fold (33.98 ± 6.7 to 16.97 ± 2.1, *p* > 0.05) and the VUS group decreased 0.7-fold (18.83 ± 3.6 to 13.08 ± 1.8, *p* = 0.0006) (Figure 3A). Also, the endogenous γH2AX levels of the PC (0.93 ± 0.18) had a 4-fold higher level when compared to the NC group (0.24 ± 0.04, *p* = 0.018). The PC also had a 1.5-fold higher level when compared to the VUS-C group (0.59 ± 0.5, *p* > 0.05) (Figure 3B). The VUS-C group (0.59 ± 0.5) had a 2.5-fold higher level when compared to the NC group (0.24 ± 0.04, *p* = 0.017). However, following Dox-induction, no statistically significant difference was observed between the groups (*p* > 0.05) (Figure 3B). Except for the PC group, which displayed a 0.4-fold decrease (0.93 ± 0.18 to 0.39 ± 0.11, *p* = 0.0173), there was no statistically significant difference between the endogenous and exogenous levels of DNA damage in the NC and VUS-C groups (Figure 3B).

DNA damage was more significant in the PC than in NCs in the sample. The variation in the PC group, which induces damage in the DNA-binding region of BRCA1 and BRCA2, provides more evidence supporting its designation as a pathogenic mutation.

### 3.5. Unveiling the Differences between VUSs

The mean % DNA in comet tail levels was not found to be significantly different between endogenous NCs (20.05 ± 3.5) and VUS-Cs (20.42 ± 3.6, *p* > 0.05), except in the case of the PC (33.98 ± 6.7, *p* > 0.05), who had approximately 2-fold % DNA in the comet tail. On the other hand, after Dox-induction, the mean of NCs increased nearly 2-fold (20.05 ± 3.5 to 32.16 ± 5.5, *p* = 0.0138), whereas the PC decreased 2-fold (33.98 ± 6.7 to 16.97 ± 2.2, *p* > 0.05) (Figure 4). In addition, after Dox-induction, NCs had approximately 2.5-fold more DNA in the comet tail of cells than VUS-Cs (Figure 4). VUS-C % DNA in the comet tail decreased after Dox-induction: VUS-C1 (T1011R) decreased 1.6-fold (20.42 ± 5.09 to 13.02 ± 1.2), VUS-C2 (T1104P/M1168K) decreased 1.4-fold (17.99 ± 7.2 to 12.96 ± 3.2), VUS-C3 (R2027K) decreased 1.5-fold (17.99 ± 0.8 to 11.98 ± 0.03), VUS-C4 (G2044A) decreased 1.5-fold (20.94 ± 2.2 to 14.14 ± 3.02), and VUS-C5 (D2819V) decreased 1.3-fold (16.81 ± 2.2 to 13.34 ± 1.8) (Figure 4).

In contrast, the phosphorylation of **γ**H2AX levels showed differences between VUS-Cs. Endogenous VUS-Cs were compared to the mean of NC subjects (0.24 ± 0.04). The PC increased 4-fold (0.92 ± 0.17, *p* < 0.0001), VUS-C1 (T1011R) decreased 0.4-fold (0.11 ± 0.01), VUS-C2 (T1104P/M1168K) increased 1.5-fold (0.35 ± 0.05), VUS-C3 (R2027K) increased 6-fold (1.38 ± 0.09, *p* < 0.0001), VUS-C4 (G2044A) increased 4-fold (0.87 ± 0.19, *p* < 0.0001), and VUS-C5 (D2819V) had no difference. After Dox-induction, the mean of NC subjects did not change significantly. The exogenous PC (0.39 ± 0.11) had no difference with the mean of exogenous NC subjects (0.33 ± 0.07) but decreased 0.4-fold when compared to the endogenous PC (*p* < 0.0001). When exogenous VUS-Cs were compared with the mean of NC subjects, VUS-C2 (T1104P/M1168K), VUS-C4 (G2044A), and VUS-C5 (D2819V) had no difference, VUS-C3 (R2027K) increased 2.3-fold (0.75 ± 0.09, *p* = 0.0150), while VUS-C1 (T1011R) (0.10 ± 0.08, *p* = 0.0203) decreased 0.3-fold after Dox-induction (Figure 5). 

### 3.6. Reclassification of the VUSs

According to the ACMG/AMP functional studies evidence codes PS3 and BS3, we classified T1011R from VUS (-) to VUS (int), T1104P from VUS (int) to VUS (+)/M1168K still VUS (−), and D2819V VUS (int) to VUS (+) with the addition of the PS3 criterion. The final classification result of VUS-C2 (T1104P/M1168K) was determined as VUS+ due to the presence of a variant with increased pathogenicity. However, R2027K and G2044A were not changed and interpreted as LB (Table 1). 

## 4. Discussion

This study delves into the functional characterization of *BRCA2* missense VUSs by assessing DNA damage responses following Dox-induction. The increasing utilization of NGS for hereditary cancer panels is acknowledged, emphasizing the growing need for accurate variant interpretation. This study aligns with the ACMG/AMP guidelines for variant classification, emphasizing the importance of functional data in resolving VUS classifications.

This study employed a comprehensive approach, combining in silico analyses with functional assays, to better understand the clinical significance of T1011R, T1104P/M1168K, R2027K, G2044A, and D2819V missense variations. We used several in silico pathogenicity prediction tools, including SIFT, PolyPhen-2, MetaRNN, DeMAG, and BayesDel, which provided initial insights into the potential pathogenicity of selected missense VUSs within the *BRCA2* gene. 

According to the in silico analysis results, T1001R, T1104P, and D2819V were supported by PP3 evidence with higher pathogenicity scores but still classified as VUSs. However, R2027K and G2044A were supported by BP4 evidence with benignity scores and classified as LB.

Recently conducted functional assays on T1011R show inconsistency; the effect on spontaneous HR T1011R was classified as HR-negative and presumed to be non-pathogenic [22]; controversially, no recombination event was observed in yeast upon expressing the T1011R variant [23]. This variant also showed impaired binding to APRIN and RAD51 proteins [24]. Mutations in the coding sequence of BRC motifs can result in the inability of BRCA2 to bind RAD51, a DNA recombinase enzyme, ultimately leading to the impairment of the HR repair [25]. It was shown that a missense mutation of a single amino acid within the sequence that encodes the BRC1, BRC2, BRC4, and BRC7 motif could prevent it from binding to RAD51 [25]. 

Dines 2020 stated that *BRCA2* exons 10 and 11 (~65% of the coding sequence) are coldspots, and of 34 P or LP missense variants in BRCA2, none are in exons 10–11. Also, more than half of reported missense VUSs in *BRCA1* and *BRCA2* are in coldspots (3115/5301 = 58.8%). Reclassifying these 3115 VUS as LB would substantially improve variant classification [26]. However, in our study, T1001R and T1104P are in exon 11 and had higher pathogenicity scores according to in silico prediction tools. 

The Clinical Genome Resource (ClinGen) Sequence Variant Interpretation (SVI) Working Group refined the PS3/BS3 criteria and qualified the homology-directed DNA repair (HDR) assay as a standardized gold standard assay [12]. The D2819V variant has been reported as non-functional in HDR assays [27].

In this study, we used comet and γH2AX assays to see how much DNA damage accumulated in the PBMCs of these VUS-Cs before and after Dox-induction. We chose Dox as a geno- and cytotoxic agent due to its wide usage in cancer treatment and ability to induce DNA damage through various mechanisms [28]. Numerous studies have extensively researched the vast array of cellular functions that are impaired by Dox, especially in the heart, and can cause cardiac failure with a poor prognosis [29]. The loss of BRCA2 in the heart led to increased DNA damage, apoptosis, and cardiac malfunction when exposed to Dox [30]. Controversially, some alkylating agents such as 0.1 μM Mechlorethamine and 1 μM Melphalan treatment in addition to 1 μM Dox decreased % tail DNA in treated BC cell lines such as MCF-7, MDA-MB 231, and T47D and demonstrated the positive effect of chemotherapeutics [31]. In addition, they showed a decreased % of tail DNA in 1 μM Cisplatin-treated colorectal cell lines, such as HCT-15, HCT-116, and HT55 [31]. Consistent with Apostolou’s results, we also found that Dox-induction decreased DSB accumulation in the PBMCs of the PC and VUS-Cs diagnosed with BC.

According to standards for the quantitative measurement of DNA damage, DNA DSBs created by anti-tumor drugs could be measured by comet and γH2AX assays [16]. Firstly, we used the alkaline comet assay to measure basal and Dox-induced DNA damage. The comet assay is a widely utilized biomonitoring instrument for determining the extent of DNA damage in human cells in response to dietary, lifestyle, environmental, and occupational exposure [32,33]. Numerous research studies have demonstrated noteworthy variations in the allocation of DNA damage among PBMCs in each individual, one of them concerning age before and after 60 [34] and another of them among smokers based on the extent of smoking [35]. The other study is that healthy nonsmoker donors’ PBMCs DNA damage increased after the 21 nM Dox-induction; however, if the cells were heat-shocked before treatment, they showed that it protected the cells from the harmful effect of the drug [36]. The results of this study indicate stability in the amount of DNA in comet tails for healthy individuals and VUS-Cs, while the PC exhibited an increase. Notably, NCs experienced a rise after Dox-induction, whereas the PC and VUS-Cs demonstrated a reduction, suggesting potential functional differences. This variability highlights the heterogeneity of VUSs and indicates that some variations may impact DNA repair mechanisms differently. Notably, the study provided evidence that VUS-Cs had levels of DNA damage like NCs before the drug. However, their DNA damage levels decreased after Dox-induction, close to the level we observed in the PC. Additionally, NC-1 vs. NC-2 shows an age-related (38 vs. 58) difference in basal and Dox-induced DNA damage between two healthy controls (Appendix A).

After the occurrence of DNA DSBs, different kinases phosphorylate H2AX and is an early event [37,38]. However, independent measures are imperative to determine the presence and type of DNA damage, as relying solely on H2AX phosphorylation can lead to misleading conclusions [39]. In our study, we acknowledged the potential limitations of using γH2AX as a sole indicator of DSBs and carefully considered this aspect in interpreting our results. Our rationale for focusing on γH2AX phosphorylation levels as a surrogate marker for DNA damage response was based on its established role in the early cellular response to DSBs and its utility as a sensitive indicator of DNA damage induction. Due to the intricate nature of DSB repair mechanisms and the consequent difficulty in identifying suitable markers for analysis, innovative fluorescence-based methods have been developed to advance the quantification of DSBs, although their interpretation can be questionable [40]. It is indeed recognized that the relationship between γH2AX levels and DSBs may exhibit variability under different experimental conditions and biological contexts, as highlighted in the literature [41]. While γH2AX is commonly used as a marker for DSBs, its quantitative correlation with the number of DSBs may be influenced by various factors, including cell type, the kinetics of DNA damage and repair processes, and additional DNA lesions [41]. Additionally, BRCA2 heterozygous missense variants may cause a decrease in γH2AX after DNA damage [42]. 

While we acknowledge the variability in the quantitative relationship between γH2AX and DSBs, it is important to note that our study aimed to assess relative changes in γH2AX phosphorylation levels within experimental conditions rather than to provide an absolute quantification of DSBs. By comparing the levels of γH2AX phosphorylation between PC, VUS-C, and NC individuals, we sought to elucidate potential differences in DNA damage response pathways associated with BRCA mutations.

In our study, γH2AX revealed distinct differences between NCs, PC, and VUS-Cs regarding DSBs and highlighted the heterogeneity of the VUS landscape. Before and after Dox treatment, an increased amount of γH2AX, representing an increased amount of DSBs, was detected in R2027K and G2044A. Based on in silico prediction results, they are tolerated, and the reason for this increase might be because these variations have intact protein functions and show normal γH2AX levels. However, T1011R, T1104P/M1168K, and D2819V express even less γH2AX protein, which may indicate deterioration in the intact protein structure. Intriguingly, a significant reduction in γH2AX expression levels was observed in the PC and VUS-Cs after Dox-induction. This reduction in DNA damage levels might indicate the effectiveness of doxorubicin in treating cells with these genetic alterations. The γH2AX assay complemented the comet assay results by evaluating the expression levels of the γH2AX protein, indicative of DNA DSBs. Notably, the presence of pathogenic variations in the PC was associated with significantly increased DNA damage, reinforcing the pathogenic nature of this subject.

At the end of the study, we combined the evidence scores and reclassified T1001R as VUS (int), T1104P/M1168K and D2819V as VUS (+), and R2027K and G2044A as LB. 

This study’s findings have several critical implications. It underscores the importance of functional assays in characterizing VUSs. While in silico analyses provide valuable initial insights, the actual impact of these variants on cellular processes can be complex and context-dependent. Functional assays, such as those employed in this study, provide a more comprehensive understanding of VUS behavior. The observed DNA damage variability among VUS carriers in % tail DNA and γH2AX expression levels emphasizes the individualized nature of these genetic variations. Not all VUSs behave alike; some may have protective effects, while others may be more detrimental. These findings highlight the need for personalized risk assessment and treatment strategies for individuals with BRCA2 VUSs.

The result discussion interprets the findings within the context of known pathogenicity and functional consequences. This study’s strength lies in incorporating both in silico predictions and functional assays, bridging the gap between genetic data and clinical relevance.

This study’s limitations, such as the small sample size and potential biases in participant selection, are acknowledged. Their significance is duly highlighted, providing transparency and a foundation for future research improvements.

## 5. Conclusions

This study contributes substantially to the ongoing efforts to understand the functional consequences of *BRCA2* missense variants. The combination of in silico predictions and functional assays, particularly in the context of Dox-induced DNA damage, sheds light on these variants’ potential clinical implications. The findings underscore the importance of individualized analyses and functional assessments in variant interpretation, paving the way for more informed clinical decisions in hereditary cancer syndromes.

## Figures and Tables

**Figure 1 genes-15-00724-f001:**
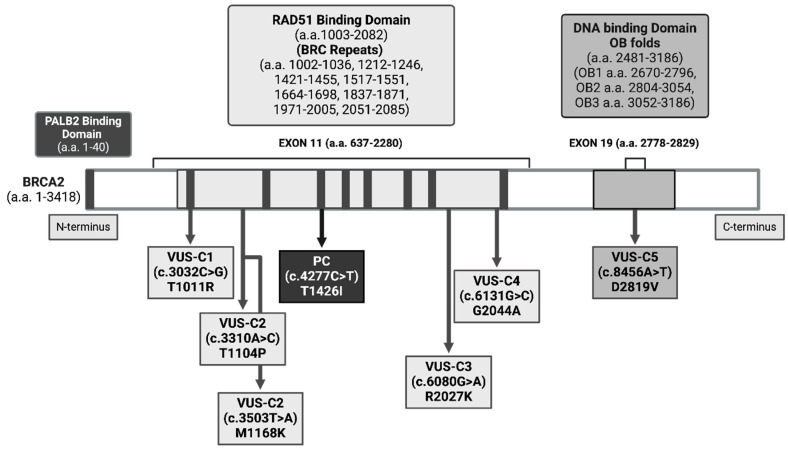
BRCA2 protein domains and the selected variants. The beginning of the N-terminus of BRCA2 is a PALB2-binding domain. The central region of BRCA2 contains a RAD51-binding domain from a.a. 1003 to 2082 and has 8 BRC repeats. This domain comprises exon 11 (a.a. 637–2280). At the C-terminus, a DNA-binding domain includes three OB folds: OB1, OB2, and OB3. The domain comprises exon 19 (a.a. 2778–2829). PC: Pathogenic carrier; BRCA1 frameshift (K711fs) deletion and a BRCA2 (T1426I) VUS carrier. VUS carriers: VUS-C1 (T1011R), VUS-C2 (T1104P) and (M1168K), VUS-C3 (R2027K), VUS-CP4 (G2044A), and VUS-C5 (D2819V).

**Figure 2 genes-15-00724-f002:**
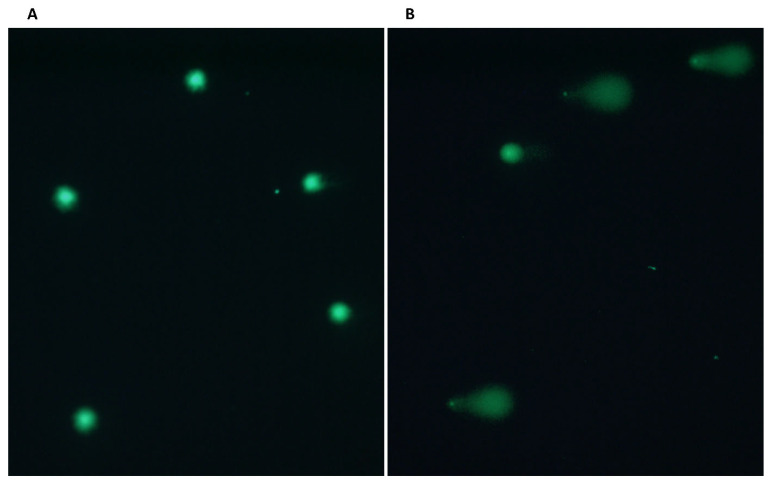
Representative comet assay results of the NC healthy control before (**A**) and after (**B**) 0.5 μM 1 h Dox-induction. Slides were analyzed, 100 cells were counted using the fluorescence microscope at ×20 magnification, and images were captured using the software. The captured cell images were scored by the CometScore program. Then, 100 cells/slide and 2 slides/person were compared. Hedgehog cells were excluded. Two single-blind researchers counted the slides.

**Figure 3 genes-15-00724-f003:**
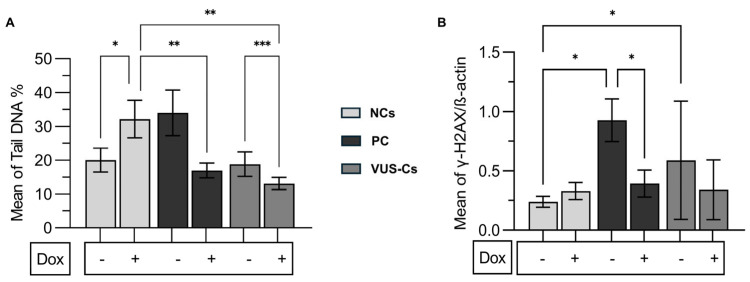
A comparison of the mean of NC healthy controls with the PC and VUS-Cs subjects before and after 0.5 μM 1 h Dox-induction. (**A**) The mean percentage of DNA in the tail before and after Dox-induction. The data are representative of two independent experiments. (**B**) H2AX phosphorylation levels before and after Dox-induction. β-actin was used as a loading control. The protein levels were normalized to the controls. The data are representative of three independent experiments. One-way ANOVA, unpaired t with Welch’s correction. Passed normality test (α = 0.05). Asterisks indicate a significant difference (* *p* < 0.05, ** *p* < 0.01, *** *p* < 0.001). Dox: doxorubicin, NCs: non-carriers, PC: Pathogenic carrier, VUS-Cs: VUS-Carriers.

**Figure 4 genes-15-00724-f004:**
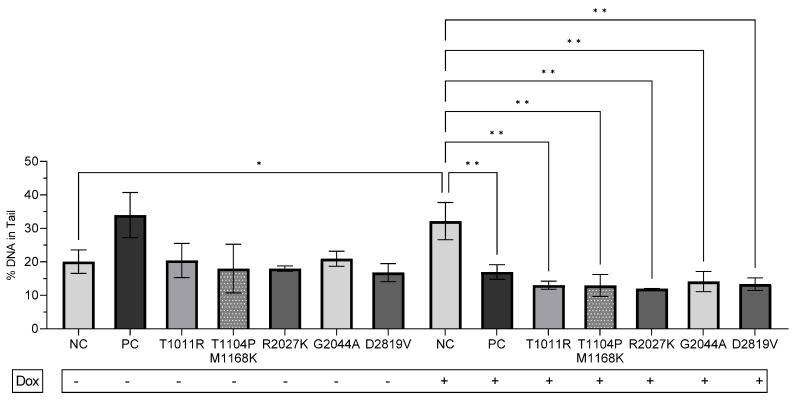
A comparison of the mean percentage of DNA in the tail before and after 0.5 μM 1 h Dox-induction between subjects. The data are representative of two independent experiments. One-way ANOVA, unpaired t with Welch’s correction. Passed normality test (α = 0.05). Asterisks indicate a significant difference (* *p* < 0.05, ** *p* < 0.01). Error bars represent the S.D. Dox: doxorubicin, NC: non-carrier, PC: Pathogenic carrier; BRCA1 frameshift (K711fs) deletion and a BRCA2 (T1426I) VUS carrier, VUS carriers: VUS-C1 (T1011R), VUS-C2 (T1104P) and (M1168K), VUS-C3 (R2027K), VUS-CP4 (G2044A), and VUS-C5 (D2819V).

**Figure 5 genes-15-00724-f005:**
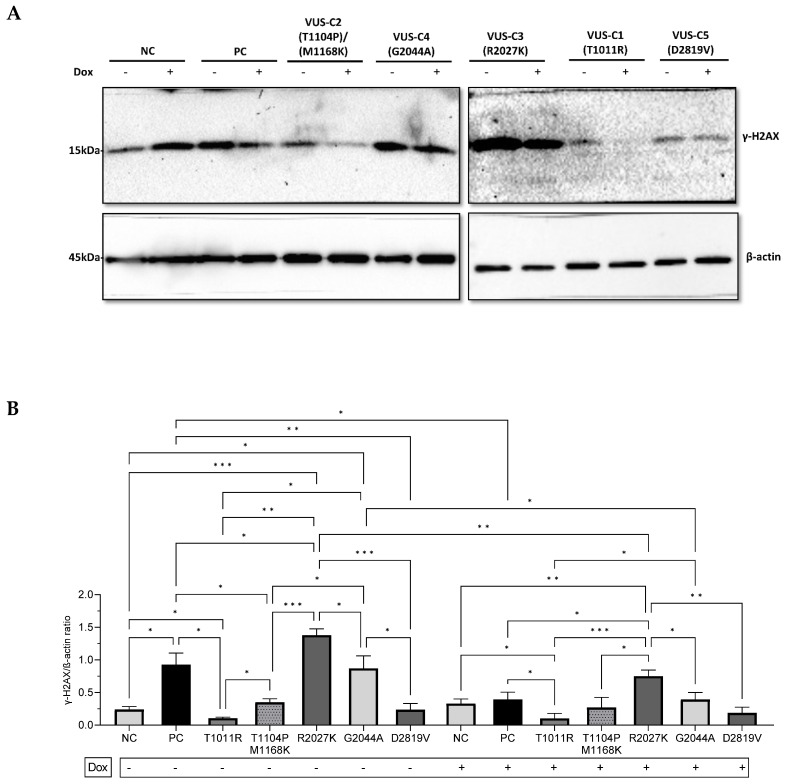
A comparison of H2AX phosphorylation levels before and after 0.5 μM 1 h Dox-induction between subjects. (**A**) A representative Western blot result showing H2AX phosphorylation levels. (**B**) Quantitative H2AX phosphorylation levels. The relative protein levels were normalized to β-actin. All experimental samples and controls were run on different gel/blots under the same conditions. All samples were studied three times independently. Dox: doxorubicin, NC: non-carrier, PC: Pathogenic carrier; BRCA1 frameshift (K711fs) deletion and a BRCA2 (T1426I) VUS carrier, VUS carriers: VUS-C1 (T1011R), VUS-C2 (T1104P) and (M1168K), VUS-C3 (R2027K), VUS-C4 (G2044A), and VUS-C5 (D2819V). Error bars represent the S.D. One-way ANOVA, unpaired t with Welch’s correction. Passed normality test (α = 0.05). The scale bar represents 0.5 μm (* *p* < 0.05, ** *p* < 0.01, *** *p*-value < 0.001).

**Table 1 genes-15-00724-t001:** In silico analysis and reclassification of BRCA2 missense VUSs.

Subjects	Gene	HGVS Coding	Mol. Con.	Protein Change	Exon	dbSNP	Clin Var	gnomAD	SIFT	PolyPhen2	MetaRNN	DeMAG	Bayes Del addAF	Bayes Del noAF	Evidence		Class	
															Previous	Recent	Previous	Recent
PC	*BRCA1*	c.2131_2132del	fs	K711fs	10	rs398122653	P	-	-	-	-	-					P	P
	*BRCA2*	c.4277C>T	ms	T1426I	11	rs748591104	CIP	1.63 × 10^−6^	0.86 T	0.001 T	0.07 T	0.05 B	−0.27 T	−0.40 T	PM2 BP6	PM2 BP6	VUS (−)	VUS (−)
	*APC*	c.3949G>C	ms	E1317Q	16	rs1801166	B/LB	-	0.02 D	0 T	0.007 T	0.12 B	-	-	-	-	-	-
VUS-C1	*BRCA2*	c.3032C>G	ms	T1011R	11	rs80358548	CIP	2.42 × 10^−5^	0 D	1 D	0.67 D	0.76 P	0.29 D	0.43 D	PM2 PP3 BP6	PM2 PP3 PS3 BP6	VUS (−)	VUS (int)
VUS-C2	*ATM*	c.4473C>T	s	F1491=	30	rs4988008	CIP	-	-	-	-	-	-	-	-	-	-	-
	*BRCA2*	c.3310A>C	ms	T1104P	11	rs80358577	CIP	7.62 × 10^−6^	0 D	1 D	0.65 D	0.64 P	0.06 T	−0.09 T	PM2	PM2 PS3	VUS (int)	VUS (+)
	*BRCA2*	c.3503T>A	ms	M1168K	11	rs80358598	CIP	7.53 × 10^−6^	0.16 T	0.006 T	0.30 T	0.08 B	−0.19 T	−0.49 T	PM2 BP4	PM2 BP4	VUS (−)	VUS (−)
VUS-C3	*BRCA2*	c.6080G>A	ms	R2027K	11	rs431825337	CIP	-	0.49 T	0.009 T	0.11 T	0.06 B	−0.23 T	−0.57 T	PM2 BP4	PM2 BP4	LB	LB
VUS-C4	*BRCA2*	c.6131G>C	ms	G2044A	11	rs56191579	CIP	1.43 × 10^−5^	0.7 T	0.062 T	0.11 T	0.06 B	−0.31 T	−0.50T	PM2 BP4 BP6	PM2 BP4 BP6	LB	LB
VUS-C5	*BRCA2*	c.8456A>T	ms	D2819V	19	rs1555287655	VUS	-	0 D	0.964 D	0.84 D	0.62 P	0.40 D	0.33 D	PM2 PP3	PM2 PP3 PS3	VUS (int)	VUS (+)

Transcript numbers are as follows: NM_007294.4 (*BRCA1*), NM_000059.4 (*BRCA2*), NM_000038.6 (*APC*), and NM_000051.4 (*ATM*). ClinVar; CIP: Conflicting interpretations of pathogenicity, VUS: variant of uncertain significance, B/LB: benign/likely benign, P: pathogenic. Mol. con.: molecular consequence; fs: frameshift, ms: missense, s: synonymous. The Sorting Intolerant From Tolerant (SIFT) score ranges from 0.0, deleterious (D), to 1.0, tolerated (T). The PolyPhen-2 (Polymorphism Phenotyping v2) score ranges from 0.0 is tolerated (T) to 1.0 is deleterious (D). MetaRNN is a deep recurrent neural network (RNN), and scores greater than or equal to 0.5 are classified as pathogenic. DeMAG (deciphering mutations in actionable genes); the higher the score, the more likely the variant is pathogenic. BayesDel is a deleteriousness meta-score. The range of the score is from −1.29334 to 0.75731. The higher the score, the more likely the variant is pathogenic. A universal cutoff value (0.0692655 with MaxAF, −0.0570105 without MaxAF) was obtained by maximizing sensitivity and specificity in classifying ClinVar variants. This database has two sets of BayesDel scores: one integrated MaxAF and one without. PS3: Well-established in vitro or in vivo functional studies supportive of a damaging effect on the gene or gene product. PM2: Absent from controls (or at extremely low frequency if recessive) in Exome Sequencing Project, 1000 Genomes Project, or Exome Aggregation Consortium. PP3: Multiple lines of computational evidence support a deleterious effect on the gene or gene product (conservation, evolutionary, splicing impact, etc.). BP4: Multiple lines of computational evidence suggest no impact on the gene or gene product (conservation, evolutionary, splicing impact, etc.). BP6: A reputable source recently reports a variant as benign, but the evidence is not available to the laboratory to perform an independent evaluation. Reclassification according to VUS (+) close to LP, VUS (−) close to LB, VUS (int) is intermediate.

## Data Availability

The data supporting this study’s findings are available from the corresponding author upon reasonable request.

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
