# Peer review of "Cosmic Whirl: Navigating the Comet Trail in DNA: H2AX Phosphorylation and the Enigma of Uncertain Significance Variants"

_genes, 2024, doi:10.3390/genes15060724_

Round 1
Reviewer 1 Report
Comments and Suggestions for Authors
The authors endeavored to elucidate the significance of BRCA2-VUSs utilizing in-silico analysis followed by wet experiments. Nonetheless, the absence of several pivotal pieces of evidence precludes me from wholeheartedly endorsing the authors' conclusion. Furthermore, certain concerns emerge from the elucidated results. At this moment, I am not able to recommend the manuscript for the publication.
Major concerns:
1. Please include one representative image each for the Comet Assay and the Western Blotting. These raw data are the critical evidences.
2. Please prepare a table for the parameters and criteria that were selected for the ACMG/AMP guided processing. For example, what is the threshold you set to define a close call between VUS to LP or LB. More details such as the raw values from the analysis should be disclosed as SI at least.
3. Did the viability of PBMCs significantly changed during the treatment of Doxorubicin? This is critical because we would lose information for dead cells.
4. I am confused about the definition of VUS-C2. It looks like a dual-substitution. However, the authors reclassified T1104P and M1168K separately.
5. It's believed that γH2AX doesn't always have a consistent quantitative connection to double-strand breaks (DSBs). I suggest the authors reinterpret the result accordingly. (https://doi.org/10.4161/cc.10.19.17448; https://doi.org/10.3390/ijms25042227)
Minor suggestions:
Move the Figure 3 to SI since it is just a detailed illustration for the Figure 4.
Author Response
Thank you very much for taking the time to review this manuscript. Please find the detailed responses below and the corresponding revisions/corrections highlighted in the re-submitted files.
Comments 1: Please include one representative image each for the Comet Assay and the Western Blotting. These raw data are the critical evidences.
Response 1: The representative images of the comet assay (Figure 2) and the western blot results were included in (Figure 5A) in the revised manuscript.
Comments 2: Please prepare a table for the parameters and criteria that were selected for the ACMG/AMP guided processing. For example, what is the threshold you set to define a close call between VUS to LP or LB. More details such as the raw values from the analysis should be disclosed as SI at least.
Response 2: Table 1 in the manuscript summarizes the parameters and criteria employed in the variant classification process, such as PS3, PM2, PP3, BP4, and BP6, used to define close calls between VUS, LP, and LB classifications. The abbreviations in the table describe the evidence codes according to ACMG/AMP criteria. The thresholds by the Varsome human genomic variant search engine were used and explained in the S1. We also shaded the table for the participants and their variants for clarity. We also added titles for the evidence codes “previous” and “recent”.
Comments 3: Did the viability of PBMCs significantly changed during the treatment of Doxorubicin? This is critical because we would lose information for dead cells.
Response 3: To ensure the integrity and reliability of our results, we carefully monitored the viability of PBMCs throughout Dox treatment in the assay optimization experiments. We employed trypan blue exclusion at the TC20 automated cell counter device (BioRad, USA) to assess cell viability and to distinguish between live and dead cells following treatment with Dox. The viability of PBMCs was not significantly affected by Dox-induction under the experimental conditions used in this study. However, we acknowledge the importance of considering the potential impact of Dox-induced cytotoxicity on the interpretation of our results, particularly concerning DNA damage response assays and downstream analyses. In light of this, we have carefully controlled for any potential confounding effects of cell viability on our experimental outcomes. Specifically, we have excluded data from samples with compromised viability to ensure the accuracy and reliability of our measurements.
Comments 4: I am confused about the definition of VUS-C2. It looks like a dual-substitution. However, the authors reclassified T1104P and M1168K separately.
Response 4: We re-classified T1104P and M1168K separately based on a nuanced evaluation of each variant's clinical significance, considering additional evidence beyond dual substitutions in the revised manuscript. Both variants may exhibit dual substitutions; their functional implications, allele frequencies, and available literature support distinct classifications (Please see also ref. 22). Re-classified variants of VUS-C2 are T1104P classified from VUS (int) to VUS (+), and M1168K didn’t change, it is still VUS (-). Upon evaluating VUS-C2, the final classification was determined as VUS+ due to the presence of a variant with increased pathogenicity, as described in the revised manuscript.
Comments 5: It's believed that γH2AX doesn't always have a consistent quantitative connection to double-strand breaks (DSBs). I suggest the authors reinterpret the result accordingly.
(https://doi.org/10.4161/cc.10.19.17448; https://doi.org/10.3390/ijms25042227)
Response 5:
We re-discussed the quantitative relationship between γH2AX and the DSBs. Together with the references that the reviewer suggested, we discussed other studies on the subject and included them in the revised manuscript (Discussion section paragraphs 9th and 10th).
The study acknowledges the potential limitations of using γH2AX as a sole indicator of DSBs and has carefully considered this aspect in interpreting the results. While acknowledging the variability in the quantitative relationship between γH2AX and DSBs, the study aimed to assess relative changes in γH2AX phosphorylation levels within experimental conditions rather than to provide absolute quantification of DSBs. By comparing the levels of γH2AX phosphorylation between PC, VUS-C, and NC individuals, the study sought to elucidate potential differences in DNA damage response pathways associated with BRCA mutations. The findings are supported by complementary assays, such as comet assay analysis, which provide additional insights into DNA damage induction and repair kinetics. Together, these complementary approaches strengthen the robustness and reliability of the study's conclusions (Ref 16, 37-43).
Comments 6: Move the Figure 3 to SI since it is just a detailed illustration for the Figure 4.
Response 6: As requested by the reviewer, Figure 3 was included in the revised supplementary files, and only Figure 4 stayed in the revised manuscript. For this reason, the order of the other figures has also changed accordingly (Please see also reviewer 2 comment 2)
Reviewer 2 Report
Comments and Suggestions for Authors
The topic of the manuscript is of great mportance on the field. The re-classification of BRCA mutations found and defined as VUS will certainly help in optimizing the treatment of patients with HBOC.
The authors analyzed retrospectically a cohort of patients with BRCA2 mutations. From these they selected 5 VUS and re-collect samples to test functional assyas able to assess more precisely a role to the deifned mutations.
The article is well structured, and the methodologies used to perfom th assay quite adequate. There are some data that sould be discussed by authors: They used comet assay and gH2Ax phospshorylation levels to define the status of the mutations. I wonder whetehr the authors have considerd the measure of H2Ax foci as a better functional assay for BRCA mutants. Several studies have found the foci well associated to HR status in both breast and ov cancer patients.
The data presented in figure 3 and 4 are redundant. They could in my opinion select one of the two, being both supporting the same conclusion.
Much attention has been given to the comet assay results while only few line to the data on gH2Ax. Perhaps a description of the finding and the reason why these data allowe or suggested a better classification should be given.
Some acronyms should be defined (for example HBOC is not explained and its definition is clear for people working in teh field only).
The authors gave sufficient emphasis on the limitations
Author Response
Thank you very much for taking the time to review this manuscript. Please find the detailed responses below and the corresponding revisions/corrections highlighted in the re-submitted files.
Comments 1: They used comet assay and gH2Ax phosphorylation levels to define the status of the mutations. I wonder whether the authors have considered the measure of H2Ax foci as a better functional assay for BRCA mutants. Several studies have found the foci well associated to HR status in both breast and ov cancer patients.
Response 1: We carefully considered various assay options when designing our study and ultimately chose western blot analysis of γH2Ax phosphorylation levels. Western blotting offers a quantitative assessment of γH2Ax phosphorylation levels, providing precise data that can be analyzed statistically to discern differences between mutant and wild-type samples. Additionally, western blotting is a widely used technique with established protocols in our laboratory, ensuring robust and reproducible results.
Comments 2: The data presented in figure 3 and 4 are redundant. They could in my opinion select one of the two, being both supporting the same conclusion.
Response 2: As requested by the reviewer, Figure 3 was included in the revised supplementary files, and only Figure 4 stayed in the revised manuscript. For this reason, the order of the other figures has also changed accordingly (Please see also reviewer 1 comment 6)
Comments 3: Much attention has been given to the comet assay results while only few line to the data on gH2Ax. Perhaps a description of the finding and the reason why these data allowe or suggested a better classification should be given.
Response 3: We did not separate the two experimental results from each other while writing our results. Upon the reviewer's recommendation, we read the manuscript carefully again. We re-discussed the quantitative relationship between γH2AX and DSBs studies and included them in the revised manuscript (Discussion section paragraphs 9th and 10th). Figure 2 consists of both Comet and H2AX results. We deemed it appropriate to include Figure 3 as a supplementary file, which repeats the same result for Comet.
Comment 4: Some acronyms should be defined (for example HBOC is not explained and its definition is clear for people working in teh field only).
Response 4: According to the reviewer's suggestion, we included the term HBOC, "Hereditary Breast and Ovarian Cancer" syndrome, in more detail and explained it in the first paragraph of the revised manuscript.
Reviewer 3 Report
Comments and Suggestions for Authors
The major focus of this manuscript is the assessing impact of inherited variation in BRCA2 protein and thereby correctly classifying previously unstudied variants. The authors used two different techniques to assess the DNA damages induced by Doxorubicin – comet assay and gamma-H2AX phosphorylation assay. They tested five Variants of uncertain significance and made interesting discovery that, while NC (no BRCA2 mutation) samples display generally low comet moment before Dox treatment and shows increase in comet moment after Dox treatment, the comet moment for PC (known significant BRCA2 mutation) sample is higher before treatment and decreases significantly after treatment. Similar pattern of decreased gamma-H2AX phosphorylation after Dox treatment was also observed for PC and VUS samples tested. Differences among the five VUSs tested were not revealed with the comet assay but with the gamma-H2AX phosphorylation assay, two of the VUS samples showed significantly more decrease post Dox treatment compared to the others. The authors suggest that these assays in combination with Dox treatment can be used to assess the significance of VUSs and reclassify them. There is an open question as to why PCs and VUSs show decreased DNA damages (in forms of gamma-H2AX phosphorylation and % DNA in comet tail) in response to Dox and whether this would be a common trend with other DNA damaging agents or specific to the Top2-poisoning/DNA-protein crosslink producing property of Dox. This could be a question that needs to be answered before adaptation of these assays in combination with other classes of DNA damaging agents. Otherwise, overall, this manuscript describes thorough and rigourous assessment of the DNA damage response in VUS BRCA2 mutant cells and report significant findings regarding a novel approach for reclassification of previously uncharacterized VUSs.
Author Response
Thank you very much for taking the time to review this manuscript. Please find the detailed responses below and the corresponding revisions/corrections highlighted in the re-submitted files.
Comments 1: There is an open question as to why PCs and VUSs show decreased DNA damages (in forms of gamma-H2AX phosphorylation and % DNA in comet tail) in response to Dox and whether this would be a common trend with other DNA damaging agents or specific to the Top2-poisoning/DNA-protein crosslink producing property of Dox. This could be a question that needs to be answered before adaptation of these assays in combination with other classes of DNA damaging agents.
Response 1: As requested by the reviewer, we explained in detail the effects of other DNA-damaging agents based on the study of Apostolou et al. (2014). The study used some alkylating agents such as 0.1 μM Mechlorethamine, 1 μM Melphalan, and 1 μM Cisplatin in addition to 1 μM Dox. The percentage of tail DNA decreased in treated cells with Dox, Mechlorethamine, and Melphalan in MCF-7 and MDA-MB 231 BC cell lines, with Dox and Melphalan in the T47D BC cell line. In addition, Cisplatin also affected some colorectal cancer cell lines such as HCT-15, HCT-116, and HT55, and the percentage of tail DNA decreased in the treated cells. All these results demonstrate the positive effect of chemotherapeutics. We answered the reviewer's question and revised it in the discussion section 7th paragraph.
Round 2
Reviewer 1 Report
Comments and Suggestions for Authors
Authors addressed most of my concerns and I have only one minor suggestion regarding the viability. Please emphasize the monitoring of viability in the manuscript so that readers won't have concern as I did.
Author Response
Thank you very much for taking the time to review this manuscript. Please find the detailed response below and the corresponding revisions/corrections highlighted in the re-submitted files.
Comment: Authors addressed most of my concerns and I have only one minor suggestion regarding the viability. Please emphasize the monitoring of viability in the manuscript so that readers won't have concern as I did.
Response: We have incorporated viability monitoring into the supplementary material and methods section of the manuscript.